# Small is beautiful, but large is certified: A comparison between fisheries the Marine Stewardship Council (MSC) features in its promotional materials and MSC-certified fisheries

Frédéric Le Manach[1]*, Jennifer L. Jacquet[2], Megan Bailey[3], Charlène Jouanneau[4], Claire Nouvian[1]

1 BLOOM, Paris, France, 2 Department of Environmental Studies, New York University, New York, NY, United States of America, 3 Marine Affairs Program, Dalhousie University, Halifax, NS, Canada, 4 Fisheries Consultant, Concarneau, France

* E-mail: fredericlemanach@bloomassociation.org

**Data Availability Statement:** All data and processing scripts are available from http://dx.doi.org/10.17632/gpynbmn7f9.1. NB: In order to

## Abstract

The Marine Stewardship Council (MSC) sets a standard by which sustainable fisheries can be assessed and eco-certified. It is one of the oldest and most well-known fisheries certifications, and an estimated 15% of global fish catch is MSC-certified. While the MSC is increasingly recognized by decision-makers as an indicator for fishery success, it is also criticized for weak standards and overly-lenient third-party certifiers. This gap between the standard's reputation and its actual implementation could be a result of how the MSC markets and promotes its brand. Here we classify MSC-certified fisheries by gear type (i.e. active vs. passive) as well as by length of the vessels involved (i.e. large scale vs. small scale; with the division between the two occurring at 12 m in overall length). We compared the MSC-certified fisheries (until 31 December 2017) to 399 photographs the MSC used in promotional materials since 2009. Results show that fisheries involving small-scale vessels and passive gears were disproportionately represented in promotional materials: 64% of promotional photographs were of passive gears, although only 40% of MSC-certified fisheries and 17% of the overall catch were caught by passive gears from 2009–2017. Similarly, 49% of the photographs featured small-scale vessels, although just 20% of MSC-certified fisheries and 7% of the overall MSC-certified catch used small-scale vessels from 2009 to 2017. The MSC disproportionately features photographs of small-scale fisheries although the catch it certifies is overwhelmingly from industrial fisheries.

## 1. Introduction

Operating outside of traditional, state-based governance frameworks for fisheries management, the Marine Stewardship Council (MSC) arose in 1997 as a means for consumers to

ensure anonymity during the review process, a Dropbox link is provided in the manuscript. It will be replaced with the Mendeley repository, should this manuscript be accepted for publication.

**Funding:** The authors received no specific funding for this work.

**Competing interests:** The authors have declared that no competing interests exist.

reform fisheries through their purchasing power [1]. Founded by the (then) largest international consumer goods company (Unilever) and the world's largest environmental NGO (WWF, then known as the World Wildlife Fund), the MSC is a private third-party certification system that sets a global standard by which "the sustainability of a fishery can be assessed regardless of its size, geography or the fishing method used" [2, 3]. Fisheries seeking entry into the program are assessed against the standard by consultants from third-party certifiers (called 'conformity assessment bodies' or 'CABs'; referred to here as 'certifiers'). Fisheries that meet the requirements of the standard are approved by the certifiers and deemed sustainable, and products from these fisheries have the option of using the MSC label (see Table 1 for details on the certification process). Based on the MSC's online database, as of 31 December 2017, 210 fisheries were certified (S1 Table), 76 suspended (i.e. the fishery was certified but then lost the

**Table 1. Summary of the certification process[a].**  Source: [4].

| Step | Description |
|---|---|
| 1 | The candidate fishery chooses a third-party certifier (e.g. Lloyds Register) and establishes a contract with it once the 'unit of assessment' is determined. The candidate fishery remunerates the certifier. |
| 2 | The certifier conducts a pre-assessment (which is optional) at the request of the candidate fishery, to assess whether certification is achievable. |
| 3 | The certifier conducts a full assessment according to the three principles of the MSC standard, which is meant to be comprehensive in assessing the ecological impacts of a fishery: impact on the target stock, impact on the ecosystem, and effectiveness of the overarching management regime.<br>To be certified, the candidate fishery must score at least 60 percent for each of the 28 'performance indicators' across the three principles. An average score of 80 must be reached for each of the principles. At the end of the process, the 'final report and determination' is issued, which states whether the fishery should be certified or not. Stakeholders can get involved at different stages of this process.[b] |
| 4 | If there is opposition to the certification, civil society groups may decide to object to the results of the determination.[c] The cost of this objection [GBP15,000 until 2010, GBP5,000 afterward; 5] is borne by the objector. In such an event, the MSC then assigns and remunerates an 'independent adjudicator' to decide whether the objection should be upheld. |
| 5 | If there was no objection or if the objection process was not successful, the fishery is certified and can use the MSC logo, on which royalties are levied.[d] |
| 6 | Annual audits are conducted by the certifier, which may result in various cases in the suspension of the certificate (e.g. negative scientific advice). The certified fishery must also re-enter the full certification process every five years. |

[a] In 2016, the MSC initiated a program to "streamline, improve stakeholder engagement, and reduce the complexity of fishery assessments", which resulted in a new 'Fisheries Certification Process' being released in August 2018 and implemented in February 2019. This streamlined process does not cover the time-period studied here and is thus not accounted for in this table. According to the MSC, "the new process aims to frontload stakeholder input into a fisheries assessment, increase the amount of meaningful input periods for stakeholders, and help to focus the third-party assessment team at site visits on the right questions, leading to more robust assessment reports" (see https://improvements.msc.org/database/streamlining).

[b] According to the MSC, the cost of a full assessment ranges from USD15,000 to USD120,000 [4].

[c] A diversity of stakeholders, including scientists, eNGOs [including its founding body WWF; 6], commercial fishers, and chefs have expressed concerns over the direction of the MSC and have objected to the certification of certain fisheries. During its first 15 years of existence, i.e. from 1997 to 2012, scientists, NGOs, and other representatives of civil society filed 32 formal objections in the certification of 30 different fisheries and only two of these objections were upheld [7].

[d] In the early 2000s, the MSC's operational costs started to be covered by annual licensing fees and royalties from companies using the label on public-facing products [tiered rate starting at 0.5% of the net wholesale value of MSC-labeled seafood sales; 8] and these fees are intended to supplement and replace the philanthropic funding the MSC has received. In 2019, they accounted for 21 million GBP of the MSC's 26 million GBP annual revenue [i.e. 80%; 9].

certificate) or withdrawn (i.e. the fishery was certified but then withdrawn from the program), while 44 had failed the MSC's initial assessment.

Due to its leading position on the labeled seafood market, the MSC has been widely studied by scholars. Some studies note the MSC's positive impact, especially by i) the empowerment of small island developing states to gain greater sovereign control over fisheries and increase generation of wealth [10–13]; ii) their rigorous standards/processes and strong market intake as a means to achieve sustainability and increase public awareness [5, 14–20]; and iii) the creation of economic value and other benefits [e.g. improved governance, reputational benefits etc.; 12, 19]. Another body of work is critical of the MSC. These papers have focused on i) outcomes favoring large-scale, independent and internationally-oriented fisheries [1, 21–27], ii) certification of overexploitation of overfished stocks [28, 29], iii) certification of invasive species [30, 31], iv) certification of key species for marine ecosystems [e.g. krill destined for natural supplements and fishmeal, sometimes fished in sensitive areas of the world with relatively minimal anthropogenic impacts, such as the Southern Ocean; 22]; and v) the subjectivity, inconsistency, and leniency of third-party certifiers and adjudicators [7, 32].

Globally, the MSC is the most visible eco-label for seafood, and smaller consumer-facing programs such as Seafood Watch now largely promote the MSC [33]. The number of MSC-certified fisheries in a country is also now included in the Convention on Biological Diversity (CBD) as an indicator for CBD Target 4 ["Governments, businesses and stakeholders at all levels have taken steps to achieve or have implemented plans for sustainable consumption"; 34] and Target 6, which aims to have all fisheries "within safe ecological limits" by 2020 [35]. According to the MSC, 15% of the world's catch was already certified in 2019 [9], and it publicly announced its aim to have one third of the world's fish catch certified or in assessment by 2030 [36]. According to the MSC, the number of products with the MSC label displayed to consumers has increased 34-fold between 2008 and 2019, reaching 40,000 products [9]. This label is not the only visual item that the MSC uses to promote itself—it also uses a wide range of visuals in its promotional materials, including in annual reports and other media of public communication such as Facebook. As with agricultural production, there is a widespread perception that 'small is beautiful' [37–40], which led us to question whether the MSC was accurately representing its certified fisheries in its promotional materials. Here we compare the fisheries the MSC used in its promotional materials (i.e. in documents such as financial reports, and on Facebook) to MSC-certified fisheries.

## 2. Materials and methods

To compare promotional materials with on-water certifications, we created two datasets that included 1) all photographs showcasing fisheries that were used in public reports published by the MSC and on Facebook since 2009, and 2) all fisheries (by catch, gear type, and scale) that have been MSC-certified until 31 December 2017.

The vessels involved in the MSC-certified fisheries were classified as either 'small scale' or 'large scale'. We used the European definition of 'small-scale, coastal fisheries': "fishing vessels of an overall length of less than 12 m and not using towed fishing gear" [i.e. bottom trawls & dredges; 41, 42]. Vessels longer that 12 m or using towed fishing gears were deemed 'large scale'.

We categorized gears in the photographs as either 'active' or 'passive' (Table 2), a distinction that is used by a wide range of fisheries actors around the world, such as the United Nations, the European Commission, national administrations (e.g. the National Oceanic and Atmospheric Administration in the USA), environmental NGOs, and fishers' representatives. 'Passive' (or 'static' gears) are often seen as a proxy for small-scale vessels [43], although this is not

**Table 2. Characteristics of 'passive' vs. 'active' gears.**

| Category | Spatial extent | Interaction with fish/marine invertebrates | Gears included |
|---|---|---|---|
| Passive (or 'static') | Deployed in a given space and subsequently left for a certain amount of time. | Caught through their own interaction with the gear. | Entangling nets, hooks & lines, pots & traps, longlines, hand-operated gears, other set gears (e.g. ropes for mussel cultivation). |
| Active (or 'mobile' or 'towed') | Engine-propelled and dragged, towed or moved along the seabed or across the water column. | Caught through the motion of the gear. | Bottom trawls & dredges (including Scottish/Danish seines), pelagic trawls, and purse seines. |

always true (e.g. many longline or pot-and-trap vessels are 'large scale' according to the definition used here, i.e. larger than 12 m). 'Passive' gears and small-scale vessels are often considered to have less of an environmental footprint (e.g. little physical impact on marine habitats, low fuel consumption) and are often promoted as the most sustainable practices by fisheries scientists [e.g. 44, 45], environmental NGOs [e.g. 46], and small-scale fishers' representatives [e.g. 47].

Table 3 describes the process used to produce and complement these two datasets. Raw data as well as processing scripts are available at: https://doi.org/10.17632/gpynbmn7f9.1.

Using these two datasets, we were able to compare the gear types (i.e. 'active' vs. 'passive') and vessel lengths (i.e. 'large scale' vs. 'small scale') of MSC-certified fisheries (in terms of both catch and number) with the fisheries represented in the MSC's promotional materials.

## 3. Results

### 3.1. Promotional materials

The 399 photographs analyzed in this paper show 95 unique MSC-certified fisheries (314 photographs) as well as a number of non-identified fisheries (85 photographs). Among the 95 MSC-certified fisheries, 57 were pictured only once or twice, and 13 over five times.

Gear-wise, fisheries involving active gears only represented 32% of the photographs, whereas fisheries involving passive gears accounted for 64% of all photographs. The remaining 4% consisted in a small set of photographs showing undetermined gears (Fig 1).

Fisheries involving large-scale and small-scale vessels were featured almost equally in the MSC's promotional materials, with 47% of photographs showing fisheries involving large-scale vessels, and 49% of photographs showing fisheries involving small-scale vessels (Fig 1). The remaining 4% correspond to photographs showing an undetermined scale. Overall, the MSC tends to use only a few fisheries involving small-scale vessels, but advertises each of them more often than fisheries involving large-scale vessels.

We also note that African fisheries were showcased in at least 37 photographs (i.e. 9%). However—besides the South African hake trawl fishery (the only MSC-certified African fishery; featured 14 times)—all other photographs represent uncertified and even never-assessed fisheries, mostly in Madagascar and Gambia (10 photographs each). Overall, fisheries that have never been certified accounted for at least 7% of all photographs we examined.

### 3.2. MSC-certified fisheries

The overall reference catch for the 210 fisheries certified as of 31 December 2017 amounted to 11.6 million tonnes, out of which 9.8 and 10.7 million tonnes originate from fisheries using active gears and operating large-scale vessels, respectively (Table 4). The 76 fisheries that are no longer MSC-certified represent an aggregate reference catch of 1.3 million tonnes.

In terms of catch of MSC-certified fisheries, active gears—i.e. bottom trawls and dredges (including Scottish/Danish seines), pelagic trawls, and purse seines—accounted for 83% of the

**Table 3. Creation of the two datasets used to compare the fisheries the MSC features in its promotional materials and MSC-certified fisheries.**

| Dataset 1: promotional materials[a] | Dataset 2: MSC-certified fisheries[b] |
|---|---|
| *Step 1: data aggregation* | |
| All reports that contained photographs of fisheries-related activities (i.e. in which either at least part of a vessel *and/or* a fisher was visible) were downloaded from the MSC's main website (www.msc.org) between May and October 2017, and an update was performed in February 2020 during the review process. This amounted to 27 reports published since 2009 (documents specific to particular fisheries—e.g. 'Community catch' newsletters—were excluded so as not to bias the analysis towards particular fisheries).[c]<br><br>In addition, photographs of fisheries-related activities were also compiled from Facebook (page of the London office). In total, 399 fisheries-related photographs were analyzed.[d]<br><br>Due to the fact that many photographs (especially from Facebook) could not be associated to a specific MSC-certified fishery—either because it showcased an undetermined or a non-certified fishery—we did not attribute a 'reference catch' (see opposite) to them. As a result, our analysis of promotional materials was only conducted in terms of number of photographs, but not in terms of catch (which is the case for Dataset 2). | The list of MSC-certified fisheries was downloaded on the MSC's fisheries portal (https://fisheries.msc.org) in January 2018.<br>Data provided on this website did not always include a 'reference catch' for the fisheries. Therefore, we collated this important information from other sources, as described in S1 Text. This step allowed us to obtain a time-series of the MSC-certified fisheries not only in 'number', but also in 'catch'. As a result, the time-series in catch gives more weight to fisheries catching large quantities of seafood, whereas the time-series in number gives the same weight (i.e. one unit) to all fisheries. |
| *Step 2: gear category* | |
| When possible, photographs were associated to a specific fishery *and/or* gear (e.g. 'Australian Western rock lobster' using 'Pots & traps'). In a few cases, photographs could thus be associated to either a specific fishery *or* to a gear, but not to both (e.g. 'Unidentified fishery' using 'Pots & traps', or 'Alaska salmon fishery' using 'Unidentified gear'). | For fisheries that used multiple fishing gears, the proportion of the catch by gear type, when available, was sourced from the most recent 'public comment reports' and 'annual surveillance reports'.<br>These proportions were assumed to be constant over time and were applied throughout. |
| *Step 3: scale* | |
| Each photograph was categorized as either 'small scale' or 'large scale' in two ways: when the picture did not suffice to visually determine the scale of the vessel, the second dataset allowed us to do so.<br>Note that, due to the EU definition of 'small scale' vessels used here, all photographs of bottoms trawls & dredges were automatically considered as 'large scale'.<br>As a result of this process, fisheries that were pictured several times may have been classified as both 'small scale' or 'large scale', depending on the gear/vessel pictured (e.g. the Alaska salmon fishery). | Based on the information available in the same 'public comment reports' and 'annual surveillance reports' as above, fisheries were classified as involving either or both 'small scale' or 'large scale' vessels on a gear by gear basis.<br>These proportions were also assumed to be constant over time and were applied throughout. |

[a] S2 Table shows the list of the photographs that were selected for this analysis, as well as their classification by status (MSC-certified or not), country, gear category (i.e. 'active' vs. 'passive') and scale of the vessels involved (i.e. 'small' vs. 'large').

[b] S1 Table shows the list of MSC-certified fisheries, as well as additional information such as their certification date, their status (e.g. certified, suspended), or the reference catch.

[c] The MSC's website was revamped afterward and these reports—most of which are no longer available online—were archived and are available on the online repository linked to this paper.

[d] Note that photographs where only seafood was visible were not included in the analysis, but photographs showing mussels growing on ropes (which was considered as a 'gear') were included. Several photographs which were selected based on the criteria above were not included in the analysis because they did not represent actual fisheries: e.g. one of snorkelers in the Caribbean [48], one of Rupert Howes—CEO of the MSC—catching a salmon [49], and two of a research vessel [50, 51].

MSC-certified catch since 2009 (i.e. the period covered by the analysis of the promotional materials), while passive gears (including entangling nets, longlines, pots & traps, and hand-operated gears) only represented 17% of the catch (Fig 2A). Over the same time-period, vessels smaller than 12 m in overall length and not operating bottom trawls or dredges—i.e. 'small scale vessels' according to the EU definition [41, 42]—only accounted for 7% of the MSC-certified catch, the other 93% being caught by large-scale vessels (Fig 2B). Noteworthy, the proportion of fisheries involving small-scale vessels in the total MSC-certified catch has steeply declined over time, from almost 70% in late 2000 to an average of 8% since 2009.

In terms of number of MSC-certified fisheries (i.e. all fisheries having the same weight regardless of their catch), active gears accounted for 60% of the MSC-certified fisheries since

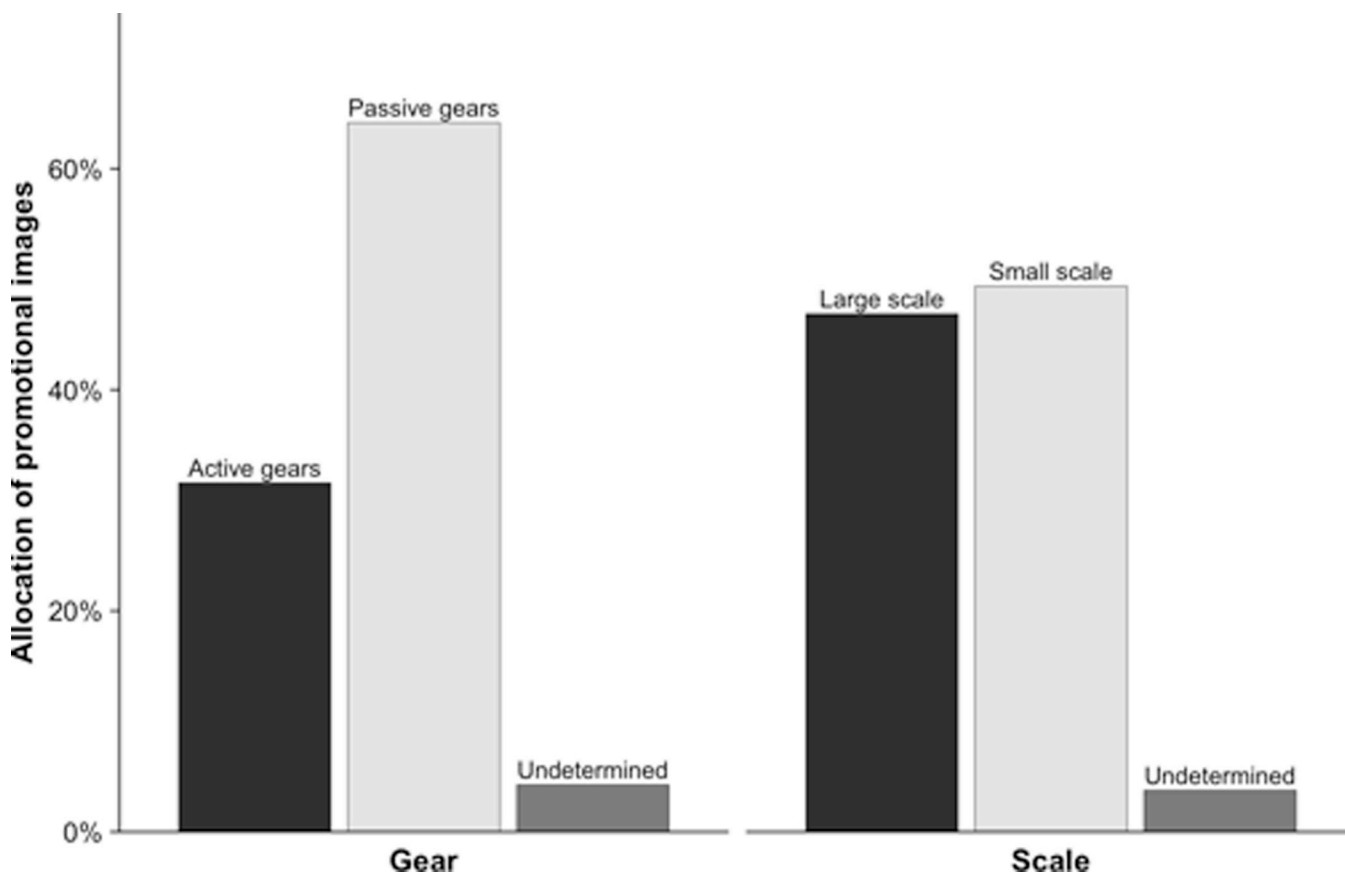

**Fig 1. Allocation of promotional images by gear (left) and scale (right), since 2009, based on 399 photographs used in MSC reports and on Facebook.**

2009, while passive gears represented 40% of the fisheries (Fig 2C); over the same time-period, small-scale vessels accounted for 20% of the MSC-certified fisheries, the other 80% being caught by large-scale vessels (Fig 2D).

## 4. Discussion

The majority of MSC-certified fisheries use active fishing gear and large-scale vessels (in either North America or Europe), which stands in stark contrast with how the MSC visually represents itself in promotional materials. Passive gears are 3.7 times (in catch; 64% vs. 17%) and 1.6 times (in number; 64% vs. 40%) more prevalent in promotional materials than are actually

**Table 4. Summary of the reference catch by gear and scale, by MSC certification status.**

| Status | Number of fisheries | Aggregate reference catch (million tonnes) | | |
|---|---|---|---|---|
| | | Overall | By gear | By scale |
| Still certified[a] | 210 | 11.6 | Active: 9.8 Passive: 1.8 | Large: 10.7 Small: 0.9 |
| Previously certified | 76 | 1.3[b] | Active: 1.0 Passive: 0.3 | Large: 1.2 Small: 0.1 |

[a] As of 31 December 2017.

[b] Last year of certification.

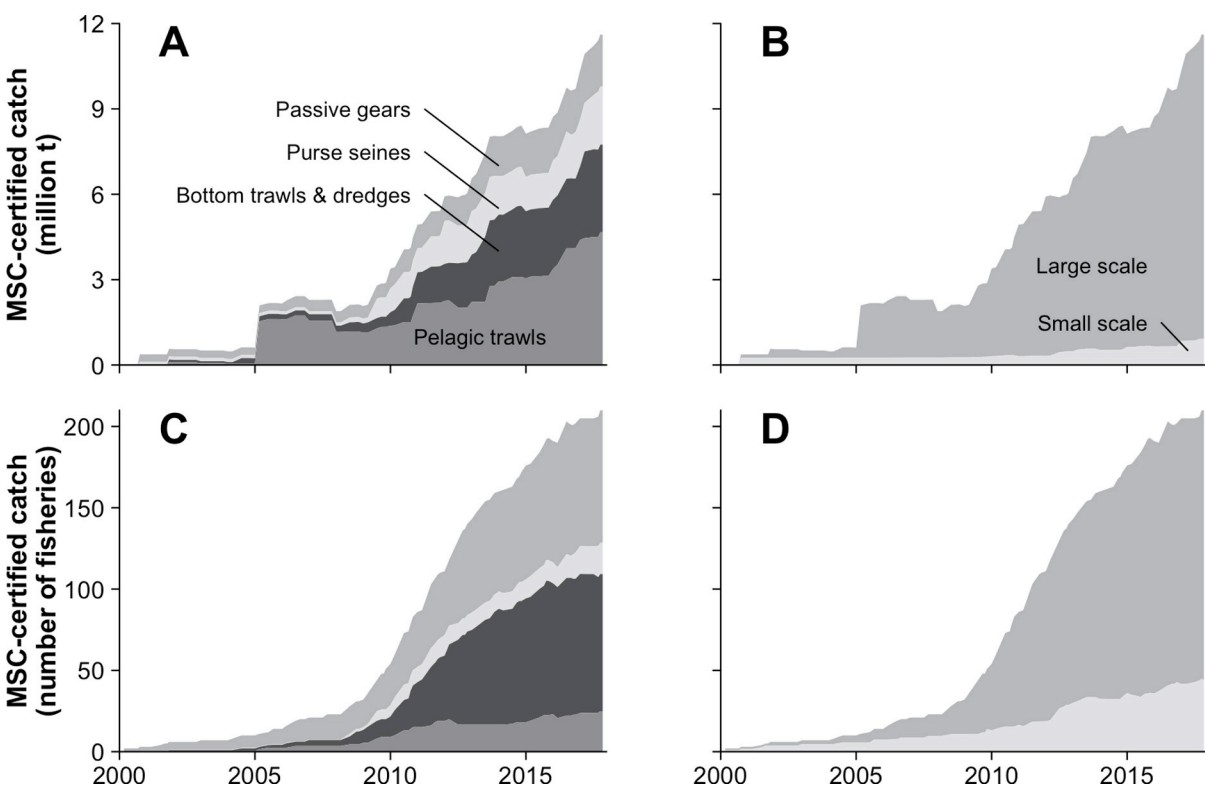

**Fig 2.** Evolution of MSC-certified catch by gear (A: in catch; C: in number) and scale (B: in catch; D: in number), 2000–2017. The 'Passive gears' category in panels A and C includes entangling nets, longlines and other hooks & lines, pots & traps, and other hand-operated gears (e.g. beach seines, hand rakes).

certified; and fisheries involving small-scale vessels 6.9 times (in catch; 49% vs. 7%) and 2.4 times (in number; 49% vs. 20%). Fisheries involving passive gears and/or small-scale vessels are therefore overrepresented in promotional materials compared to the fisheries that are actually MSC-certified, and the MSC often uses photographs of fisheries that have never been certified.

However, our results also show that the photographs used by the MSC to illustrate specific fisheries are consistent with the text: they correspond to the fisheries the MSC describes in the text, e.g. octopus fishers in Madagascar to correctly illustrate small-scale fisheries in developing countries; a large trawler to correctly illustrate the Alaska pollock fishery etc. In other words, the use of photographs of small-scale fisheries are often used alongside text about the need to increase the number of certified small-scale fisheries, especially from lower income countries. The issue we have identified is that the visual representation chosen for reports and on Facebook mostly showcase small-scale, passive fisheries, whereas mostly large-scale, active fisheries are MSC-certified.

Because sustainability is increasingly focused on climate change, we also note that fisheries involving large-scale vessels and using active gears (i.e. the majority of MSC-certified fisheries) almost always consume more fuel per unit of catch, compared to fisheries involving small-scale vessels and using passive gears [45, 52]. Research suggests that, for large-scale vessels, the most fuel-efficient fisheries are those targeting small pelagics with purse seines, and the least fuel-efficient fisheries are bottom trawlers and pot-fishing crustaceans [53, 54]. Not only does the MSC disproportionately use photographs of small-scale fisheries relative to the seafood that bears its logo, but a substantial number of the photographs feature hand-operated gears

(54 photographs, i.e. 14%), which involve the least amount of fossil fuel. In contrast, hand-operated gears only accounted for 0.3% of the MSC-certified catch since 2009.

Our analysis solely focused on reports published by the MSC as well as photographs used to promote the label on its international Facebook page, but further anecdotal evidence suggest that the same trend exists with other media. One such example is the MSC's fisheries portal (https://fisheries.msc.org), on which all documentation related to assessments and certifications can be found. On this portal, the only fisheries in the 'in focus' section (a visual carrousel showcasing MSC-certified fisheries) between May 2017 to March 2020 have been three small-scale ones using passive gears: the 'South Australia Lakes and Coorong pipi' fishery (hand-operated gears), the 'Normandy and Jersey lobster' fishery (pots & traps), and the 'Mexico Baja California red rock lobster' fishery (pots & traps). All together, these three fisheries account for 0.02% of the overall 2009–2017 MSC-certified catch. The overall catch of hand-operated gears and pots & traps—the two gears used by the three fisheries 'in focus'—accounted for 3% of the MSC-certified catch over the same time-period. As highlighted above, these small-scale, passive gear fisheries do not generate much revenue for the MSC through royalties, but they do appeal to the consumers' idealization of fisheries and appear to be consistently used by the MSC to promote its brand.

The MSC has relied on fisheries involving large-scale and/or active, higher impact fishing gears to rapidly oversee the certification of a significant part of the world's fisheries and thus become the front-line player in the sustainable seafood market. We believe that, should the MSC want to reach its target to have 30% of the World's catch either certified or in assessment by 2030, it would need to keep certifying large, industrial fisheries operating active gears. Incidentally, this would also allow the MSC to draw in higher revenues, given that a fishery such as the Western Asturias octopus fishery—which catches around 40 tonnes annually [55]—does not generate as many royalties for the MSC as the Alaska pollock fisheries, which catch around 1.5 million tonnes annually [56, 57].

Based on our findings, it appears that the MSC strongly appeals to the idealization of fisheries by consumers and policy-makers by promoting fisheries involving small-scale gears and passive gears in much higher proportions than in reality, as is the case in other sectors such as agriculture [37–39]. The MSC may be trying to appeal to the needs and desires of its consumers, providing them with the symbolic satisfaction of not having harmed the environment [58]. This strategy is already known in other food-production sectors, such as cattle farming, where consumers subconsciously perceive farms as places of "harmony and kindness" [59].

However, there is a credible risk of misunderstanding for those who scroll quickly through their documents and websites, which may explain to some extent the positive public image of the MSC. The MSC reports that 86% of consumers who know the MSC label currently trust it [60], however the risks to consumer trust associated with misleading advertising should not be ignored [61]. To maintain trust with the public, the MSC could be proactive and careful to report in a high-profile and even visual way the percentage of product coming from large-scale versus small-scale fisheries so as to ensure accurate communication to casual readers.

While the vast majority of MSC-certified catch comes from large-scale fisheries, the MSC's communication strategy focuses on small-scale, lower impact fisheries. We hypothesize that this discrepancy between MSC-certified fisheries and what the MSC advertises aims to 'green' its image with consumers. We further posit that this discrepancy might be the reason behind a perceived gap between its 'supporters'—including policy-makers—and other stakeholders that have gradually disengaged or become critical of the MSC (e.g., coalitions such as *On The Hook* or *Make Stewardship Count*). 'Small is beautiful' and the MSC favors representations of pastoral fisheries in its promotional materials, but large-scale, active fisheries represent the majority of MSC-certified fisheries.

## Supporting information

**S1 Text. Description of the process used to produce a time-series of the MSC-certified catch.**
(DOCX)

**S1 Table. Summary of the MSC-certified fisheries, as of 31 December 2017 (ordered by certification date within each status category).**
(DOC)

**S2 Table. Summary of the pictures used in the analysis, 2009–2017.**
(DOCX)

## Acknowledgments

We gratefully acknowledge Rainer Froese and Laurenne Schiller for their insightful comments and edits on various drafts of the manuscript, as well as the insightful comments and suggestions from anonymous reviewers.

## Author Contributions

**Conceptualization:** Frédéric Le Manach, Jennifer L. Jacquet, Charlène Jouanneau, Claire Nouvian.

**Data curation:** Frédéric Le Manach.

**Formal analysis:** Frédéric Le Manach, Jennifer L. Jacquet, Megan Bailey, Charlène Jouanneau, Claire Nouvian.

**Investigation:** Frédéric Le Manach.

**Methodology:** Frédéric Le Manach, Jennifer L. Jacquet, Claire Nouvian.

**Software:** Frédéric Le Manach.

**Supervision:** Frédéric Le Manach, Claire Nouvian.

**Validation:** Frédéric Le Manach, Jennifer L. Jacquet, Megan Bailey, Claire Nouvian.

**Visualization:** Frédéric Le Manach, Jennifer L. Jacquet.

**Writing – original draft:** Frédéric Le Manach, Jennifer L. Jacquet, Megan Bailey, Charlène Jouanneau, Claire Nouvian.

**Writing – review & editing:** Frédéric Le Manach, Jennifer L. Jacquet, Megan Bailey, Charlène Jouanneau.

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
