## [Decision Letter · Decision Letter 0]

24 Jan 2020

PONE-D-19-34289

Small is beautiful, but large is certified: a comparison between fisheries the Marine Stewardship Council (MSC) features in its promotional materials and MSC-certified fisheries

PLOS ONE

Dear Dr. Le Manach,

Thank you for submitting your manuscript to PLOS ONE. After careful consideration, we feel that it has merit but does not fully meet PLOS ONE’s publication criteria as it currently stands. Therefore, we invite you to submit a revised version of the manuscript that addresses the points raised during the review process.

Before this ms can be published, the authors need to address these points:

1. One of the reviewers raised an issue about the paper’s tone and conclusion that the MSC is deliberately misleading to "green" its image.  In particular, because the MSC reports also contain many stories and images of large scale industrial fisheries, concluding there is a deliberate attempt to mislead would require  demonstrating that the focus or portrayal of the stories themselves is driving misunderstandings among readers, rather than just a difference in the % of images from a certain type of fisheries versus the % share by volume those fisheries represent in the certified total.  Thus, the reviewer suggests that the current analyses support a conclusion along the lines that there is at least a credible risk of misunderstanding from casual readers, and that the MSC should be proactive and careful to report in a high profile way the percentage of product coming from different types of fisheries and regions, to ensure casual readers are indeed not mislead.  I support this assessment and urge the authors to revise the wording of their conclusions, and to include the aforementioned recommendation..

2. The other reviewer objected to the use of the word “pristine” or even “relatively pristine” when describing the Southern Ocean sector where the krill fishery occurs. I urge the authors to consider the reviewer’s comments and to modify the text accordingly.

3. Define the term “volume”, the first time it is used, or in Table S1. Explicitly explain if this term is the same as tonnage? In particular, Table S2 uses the term “tonnage”.

4. Please address the question of the implication (and meaning) of assigning fisheries to continents. In particular, explain how the results would differ when considering EEZs / ports of origin.  The reviewer raised a specific question about the Antarctic toothfish fishery and the Antarctic krill fishery, which are pursued by international vessels in a distant continent.  It would be great if the authors could discuss the implications of using these different methodological approaches on this fishery.   

5. Finally, concerning the discussion about MSC’s ‘green washing, it would be interesting to discuss the amount of fuel needed for various high-seas fisheries (and CO2 emissions), and how these aspects could be incorporated into fishery certifications.

We would appreciate receiving your revised manuscript by Mar 09 2020 11:59PM. To enhance the reproducibility of your results, we recommend that if applicable you deposit your laboratory protocols in protocols.io, where a protocol can be assigned its own identifier (DOI) such that it can be cited independently in the future. For instructions see: http://journals.plos.org/plosone/s/submission-guidelines#loc-laboratory-protocols

We look forward to receiving your revised manuscript.

Kind regards,

David Hyrenbach, Ph.D.

Academic Editor

PLOS ONE

Journal Requirements:

Reviewers' comments:

Reviewer's Responses to Questions

**Comments to the Author**

1. Is the manuscript technically sound, and do the data support the conclusions?

Reviewer #1: Yes

Reviewer #2: No

2. Has the statistical analysis been performed appropriately and rigorously? 

Reviewer #1: Yes

Reviewer #2: No

3. Have the authors made all data underlying the findings in their manuscript fully available?

Reviewer #1: Yes

Reviewer #2: Yes

4. Is the manuscript presented in an intelligible fashion and written in standard English?

Reviewer #1: Yes

Reviewer #2: Yes

5. Review Comments to the Author

Reviewer #1: Great paper, requiring a huge amount of effort. Well put together, and easy to read. I had a few little comments that maybe authors would find useful.

Ln 11 ff, throughout paper. I don’t understand the use of the word ‘volume’. Might you give a definition the first time it is used, or in Table S1? Is this the same as tonnage? In Table S2, you use tonnage.

Ln 68. I object to the use of the word ‘pristine’ or even ‘relatively pristine’ when describing the Southern Ocean and particularly that part of it where the Antarctic krill fishery occurs. ‘relatively pristine’ is an oxymoron, but how is this area (FAO 48.1) even remotely pristine having all its ground fish fished to economic ruin (fisheries now closed by CCAMLR), a million whales removed (now recovering, though a long way to go), and fur seals and elephant seals decimated (now recovered). Granted there is no development of anoxic zones owing to pollution run-off, no major oil spills, and no pollution from plastic that degrade other oceans. Maybe the term to use to describe this area is ‘relatively minimal anthropogenic influence’ (i.e. Halpern et al. 2008, though Halpern et al. admit (2009) that they didn’t go back in time far enough in their analysis to include the extirpation of fish and marine mammals from this area).

Ln 71. And because of MSC’s marketing power, smaller certifiers such as Seafood Watch and Fish-wise have thrown in the towel and now pretty much agree with MSC on its evaluations.

Ln 83. certified fisheries

Ln 111-113. I don’t know the degree to which MSC has certified open-ocean fisheries, other than Antarctic toothfish, but assigning fisheries to these continents means, what, you are just reviewing those fisheries within various EEZs (200 nm of continents?)? Or do these continents somewhat represent port of origin (well, except Antarctica)? You could maybe refer to FAO areas? Well, reading on in ‘Step 4: continent’ of table you apparently are using port of origin. I’d be interested in how Antarctic toothfish would be viewed by your scheme (fishery in FAO 88.1 and 88.2 being certified), those fish being taken by vessels from NZ, UK, S Korea, Ukraine and sometimes others.

Ln 155. 49% of photographs

Ln 206-212. Curious about the Antarctic toothfish fishery in FAO 88.1, 88.2. Maybe not the highest in terms of catch tonnage, but certainly one of the highest in terms of monetary value. What is it’s ‘origin’? Or for that matter, Antarctic krill fishery, though that is mostly Norway these days (or at least Norwegian vessels got certified). In terms of MSC’s ‘green washing,' though it would be a huge task, and I’m not asking you that it be done, the amount of power and fuel needed for various fisheries (and CO2 emmissions), especially high-seas, remote ones, would maybe be something that consumers these days would find of great interest???? Maybe somewhere in the paper you could mention that ‘active’ fisheries produce a lot of greenhouse gas emmissions, especially fisheries pulling large nets requiring a lot of horsepower??? Maybe you mentioned this, and I missed it.

Ln 260. Isn’t this a bit of dreaming? Who among those commanding a large audience is going to lead the push for MSC being honest? WWF?

Ln 276. Delete comma

Table S2. I noticed inconsistent capitalization in the names of fisheries.

Reviewer #2: Most of the MSC reports analyzed contain stories on different fisheries and issues, including a long-running and growing theme on the need to increase the number of certified small scale fisheries, especially from lower income countries. Those stories are appropriately illustrated with images from such small scale fisheries. The method does not account for this, and as a result the paper may be overreaching by concluding this is a deliberate misleading attempt to "green" the MSC image. These reports also contain many stories and images of large scale industrial fisheries - the MSC is not hiding these from readers - undermining the conclusion further. To conclude there is a deliberate attempt to mislead, the paper would need to do more I feel to demonstrate that the balance of the stories themselves is driving misunderstandings among readers, rather than just a difference in the % of images from a certain type of fisheries versus the % share by volume those fisheries represent in the certified total. The method "as is" may perhaps support a conclusion along the lines that there is at least a credible risk of misunderstanding from casual readers, and that the MSC should be proactive and careful to report in a high profile way the percentage of product coming from different types of fisheries and regions, to ensure casual readers are indeed not mislead.

6. PLOS authors have the option to publish the peer review history of their article (what does this mean?). If published, this will include your full peer review and any attached files.

Reviewer #1: No

Reviewer #2: Yes: Jim Cannon

---

## [Author Response · Author response to Decision Letter 0]

9 Mar 2020

COMMENTS FROM THE EDITOR 

1. One of the reviewers raised an issue about the paper’s tone and conclusion that the MSC is deliberately misleading to "green" its image. In particular, because the MSC reports also contain many stories and images of large-scale industrial fisheries, concluding there is a deliberate attempt to mislead would require demonstrating that the focus or portrayal of the stories themselves is driving misunderstandings among readers, rather than just a difference in the % of images from a certain type of fisheries versus the % share by volume those fisheries represent in the certified total. Thus, the reviewer suggests that the current analyses support a conclusion along the lines that there is at least a credible risk of misunderstanding from casual readers, and that the MSC should be proactive and careful to report in a high-profile way the percentage of product coming from different types of fisheries and regions, to ensure casual readers are indeed not mislead. I support this assessment and urge the authors to revise the wording of their conclusions, and to include the aforementioned recommendation.

2. The other reviewer objected to the use of the word “pristine” or even “relatively pristine” when describing the Southern Ocean sector where the krill fishery occurs. I urge the authors to consider the reviewer’s comments and to modify the text accordingly.

3. Define the term “volume”, the first time it is used, or in Table S1. Explicitly explain if this term is the same as tonnage? In particular, Table S2 uses the term “tonnage”.

4. Please address the question of the implication (and meaning) of assigning fisheries to continents. In particular, explain how the results would differ when considering EEZs / ports of origin. The reviewer raised a specific question about the Antarctic toothfish fishery and the Antarctic krill fishery, which are pursued by international vessels in a distant continent. It would be great if the authors could discuss the implications of using these different methodological approaches on this fishery. 

5. Finally, concerning the discussion about MSC’s ‘green washing, it would be interesting to discuss the amount of fuel needed for various high-seas fisheries (and CO2 emissions), and how these aspects could be incorporated into fishery certifications.

Thank you for this summary of suggested revisions. Please note that we address all of them in details below.

 

COMMENTS MADE BY REVIEWER #1

1. Ln 11 ff, throughout paper. I don’t understand the use of the word ‘volume’. Might you give a definition the first time it is used, or in Table S1? Is this the same as tonnage? In Table S2, you use tonnage.

Thank you for pointing out this confusion. We replaced the terms ‘volume’ and ‘tonnage’ throughout the manuscript with ‘catch’ (and specified ‘tonnes’ when appropriate).

2. Ln 68. I object to the use of the word ‘pristine’ or even ‘relatively pristine’ when describing the Southern Ocean and particularly that part of it where the Antarctic krill fishery occurs. ‘relatively pristine’ is an oxymoron, but how is this area (FAO 48.1) even remotely pristine having all its ground fish fished to economic ruin (fisheries now closed by CCAMLR), a million whales removed (now recovering, though a long way to go), and fur seals and elephant seals decimated (now recovered). Granted there is no development of anoxic zones owing to pollution run-off, no major oil spills, and no pollution from plastic that degrade other oceans. Maybe the term to use to describe this area is ‘relatively minimal anthropogenic influence’ (i.e. Halpern et al. 2008, though Halpern et al. admit (2009) that they didn’t go back in time far enough in their analysis to include the extirpation of fish and marine mammals from this area).

Thank you. We agree and have rephrased to replace ‘pristine’ with ‘relatively minimal anthropogenic impacts’.

3. Ln 71. And because of MSC’s marketing power, smaller certifiers such as Seafood Watch and Fish-wise have thrown in the towel and now pretty much agree with MSC on its evaluations.

Thank you. We have added this sentence: “Smaller consumer-facing programs such as Seafood Watch now largely agree with the MSC and its outcome.”

4. Ln 83. certified fisheries

Thank you for pointing out this typo. Now fixed.

5. Ln 111-113. I don’t know the degree to which MSC has certified open-ocean fisheries, other than Antarctic toothfish, but assigning fisheries to these continents means, what, you are just reviewing those fisheries within various EEZs (200 nm of continents?)? Or do these continents somewhat represent port of origin (well, except Antarctica)? You could maybe refer to FAO areas? Well, reading on in ‘Step 4: continent’ of table you apparently are using port of origin. I’d be interested in how Antarctic toothfish would be viewed by your scheme (fishery in FAO 88.1 and 88.2 being certified), those fish being taken by vessels from NZ, UK, S Korea, Ukraine and sometimes others.

Thank you for pointing this out. Due to the difficulty of assigning flag state or EEZ to MSC-certified fisheries, we have decided to remove the discussion on continents and kept only the part of the discussion on the many pictures from non-certified fisheries in Africa. These difficulties are summarized in the two following points:

i) A few massive fisheries have very large spatial footprints (e.g. PNA tuna fishery, around 800 kt; MINSA mackerel fishery, around 700kt; etc.) and we don’t have enough granularity to know how much was taken in EEZ X or Y, or even in which FAO area;

ii) Very large fisheries such as PNA and MINSA also include vessels flagged to multiple countries, sometimes with flags of convenience. Again, we do not have precise-enough data to produce a good estimate.

With regards to your specific question on toothfish fisheries, most of them actually involve vessels from only one country, e.g. France for the SARPC fishery, Australia for the HIMI fishery etc. As a result, their classification was straightforward, but in contrast, we considered that the ‘Ross Sea toothfish longline’ fishery — with vessels from New Zealand, Australia, UK, Norway, and Spain — had an ‘Undetermined’ origin for the reasons explained above.

While we have removed the part on the ‘continents of origin’, we deemed the discussion on the many pictures from non-certified fisheries in Africa still pertinent, because it shows that the MSC is eager to communicate on small-scale fisheries in developing countries, although the only MSC-certified fishery in Africa is a large-scale fishery in South Africa.

6. Ln 155. 49% of photographs

Thank you for pointing out this typo. Now fixed.

7. Ln 206-212. Curious about the Antarctic toothfish fishery in FAO 88.1, 88.2. Maybe not the highest in terms of catch tonnage, but certainly one of the highest in terms of monetary value. What is its ‘origin’? Or for that matter, Antarctic krill fishery, though that is mostly Norway these days (or at least Norwegian vessels got certified). In terms of MSC’s ‘green washing,' though it would be a huge task, and I’m not asking you that it be done, the amount of power and fuel needed for various fisheries (and CO2 emissions), especially high-seas, remote ones, would maybe be something that consumers these days would find of great interest???? Maybe somewhere in the paper you could mention that ‘active’ fisheries produce a lot of greenhouse gas emissions, especially fisheries pulling large nets requiring a lot of horsepower??? Maybe you mentioned this, and I missed it.

Thank you for this. Even though we have removed the parts of our manuscript focusing on the area of origin (see comment 5), we are now noting in our initial discussion of industrial fishing/gear types that these almost always have higher fuel use and therefore greenhouse gas emissions than the small-scale/passive gears. 

8. Ln 260. Isn’t this a bit of dreaming? Who among those commanding a large audience is going to lead the push for MSC being honest? WWF?

Thank you for this comment and your frankness. We agree that the MSC is not going to suddenly become honest if no one strongly pushes for this to happen, but the MSC has been shown to respond to criticism (e.g., their recent decision on compartmentalization). We hope that our paper will contribute to a change in the MSC’s behaviour, so as to advance the conservation of marine ecosystems. We have highly modified the discussion, as well as removed this sentence.

9. Ln 276. Delete comma

Done, thank you. 

10. Table S2. I noticed inconsistent capitalization in the names of fisheries.

Thank you. Now fixed.

COMMENTS MADE BY REVIEWER #2

1. Most of the MSC reports analysed contain stories on different fisheries and issues, including a long-running and growing theme on the need to increase the number of certified small-scale fisheries, especially from lower income countries. Those stories are appropriately illustrated with images from such small-scale fisheries. The method does not account for this, and as a result the paper may be overreaching by concluding this is a deliberate misleading attempt to "green" the MSC image. 

Thank you for this comment, which we have addressed in closer detail and have removed any language related to ‘deliberate’, which we do not have the evidence to confirm. We have added the following paragraph in the discussion: “However, our results also show that the photographs used by the MSC to illustrate specific fisheries are consistent with the text: they correspond to the fisheries the MSC describes in the text, e.g. octopus fishers in Madagascar to correctly illustrate small-scale fisheries in developing countries; a large trawler to correctly illustrate the Alaska pollock fishery etc. In other words, the use of photographs of small-scale fisheries are often used alongside text about the need to increase the number of certified small-scale fisheries, especially from lower income countries. The issue we have identified is that the visual representation chosen for these reports mostly showcase small-scale, passive fisheries, whereas mostly large-scale, active fisheries are MSC-certified". 

2. These reports also contain many stories and images of large-scale industrial fisheries - the MSC is not hiding these from readers - undermining the conclusion further. 

Thank you for this comment. It is true that there are many photos of large-scale industrial fisheries, and we have now emphasized this point in the discussion and conclusion, but we have also highly modified the discussion in a way that we hope satisfies these concerns (see below).

3. To conclude there is a deliberate attempt to mislead, the paper would need to do more I feel to demonstrate that the balance of the stories themselves is driving misunderstandings among readers, rather than just a difference in the % of images from a certain type of fisheries versus the % share by volume those fisheries represent in the certified total. The method "as is" may perhaps support a conclusion along the lines that there is at least a credible risk of misunderstanding from casual readers, and that the MSC should be proactive and careful to report in a high-profile way the percentage of product coming from different types of fisheries and regions, to ensure casual readers are indeed not mislead.

Thank you for this feedback. We have highly modified the discussion to get rid of any suggestions about “deliberate” attempts, particularly given the text does match the photographs. We have also removed most of the text about misleading advertising and greenwashing. We have deleted the entire section on ‘deliberate strategy’ and combined all of our concluding remarks into a ‘discussion’ section.

For further evidence of the disproportionate use of photographs of small-scale fisheries. We have:

1) Updated our list of reports with the 2017-18 and 2018-19 activity reports;

2) Included +100 photographs used by the MSC on its Facebook profile (London office);

3) Added a paragraph on other media, using the example of the MSC's fisheries portal (www.fisheries.msc.org). 

This allowed us to widen the scope of our analysis, as we increased the number of studied photographs from 277 to 399. Our initial results are strengthened, and the scope of the study widened, given that the categories of pictures used in reports and in Facebook are very similar with regards to vessel length and fishing gear. We have also added this in to the main text.

We very much hope this conveys our respect and appreciation for this feedback.

---

## [Editor Report · Decision Letter 1]

17 Mar 2020

Small is beautiful, but large is certified: a comparison between fisheries the Marine Stewardship Council (MSC) features in its promotional materials and MSC-certified fisheries

PONE-D-19-34289R1

Dear Dr. Le Manach,

We are pleased to inform you that your manuscript has been judged scientifically suitable for publication and will be formally accepted for publication once it complies with all outstanding technical requirements.

With kind regards,

David Hyrenbach, Ph.D.

Academic Editor

PLOS ONE
---

## [Editor Report · Acceptance letter]

21 Apr 2020

PONE-D-19-34289R1 

Small is beautiful, but large is certified: a comparison between fisheries the Marine Stewardship Council (MSC) features in its promotional materials and MSC-certified fisheries 

Dear Dr. Le Manach:

I am pleased to inform you that your manuscript has been deemed suitable for publication in PLOS ONE. Congratulations! Your manuscript is now with our production department. 

With kind regards,

on behalf of

Dr. David Hyrenbach 

Academic Editor

PLOS ONE